# Hydrophilic–Hydrophobic Properties of the Surface of Modified Carbonate Fillers for Asphalt

**Mikhail Lebedev** [1,2,*], **Valentina Yadykina** [3], **Andrey Akimov** [3], **Marina Kozhukhova** [4] and **Ekaterina Kuznetsova** [3]

1   Scientific and Educational Center "Additive Technologies", National Research Tomsk State University, Lenin Avenue, 36, 634050 Tomsk, Russia
2   Research Laboratory of the Theoretical and Applied Chemistry Department, Belgorod State Technological University Named after V.G. Shukhov, Kostukov Str., 46, 308012 Belgorod, Russia
3   Department of Automobile Roads and Railroads, Belgorod State Technological University Named after V.G. Shukhov, Kostukov Str., 46, 308012 Belgorod, Russia; vvya@intbel.ru (V.Y.); akimov548@gmail.com (A.A.); kuznecova.k@inbox.ru (E.K.)
4   Department of Civil and Environmental Engineering, University of Wisconsin-Milwaukee, 3200 N Cramer St., Milwaukee, WI 53211, USA; kozhuhovamarina@yandex.ru
*   Correspondence: michaell1987@yandex.ru

**Abstract:** The physicochemical modification of the filler allows changing the hydrophilic–hydrophobic properties and effectively influencing the processes occurring at the filler–binder interface, on which the physicomechanical characteristics of composites largely depend. The paper presents studies related to the modification of limestone-based filler effect on the degree of its hydrophobicity and wetting with liquids of different polarity, establishing the relationship between the characteristics of hydrophobized mineral powders and the adsorption capacity in relation to water. Using mechanochemical processing with hydrophobic components GF-1 and GF-2, it was possible to obtain fillers with a sufficiently high content of hydrophobic particles (58.2% and 85.9%, respectively). It was found that the results of the contact angle (123.6° and 114.5°, respectively) and the degree of hydrophobicity do not quite correlate with each other. It was noticed that the contact angle on the powder modified with GF-1 decreases with time. Studies of the powders' thermal effects wetting of different polarity liquids via microcalorimetry allows us to establish that with an increase in the filler hydrophobicity degree, the integral heat of immersion decreases due to a significant decrease in the probability of chemical interactions between water and powder due to the adsorption of applied surfactants molecules on the limestone active centers. The revealed endothermic effects indicate the occurrence of physical interactions due to non-polar dispersion forces. Differences in the nature of heat release and heat absorption in modified fillers indicate significant differences in the composition and mechanism of action of the used surfactants, which affected the efficiency of hydrophobization. At the same time, a linear dependence of the moisture absorption and moisture indicators, determined by independent experiments, on the degree of hydrophobicity was established.

**Keywords:** hydrophobization; limestone-based fillers; degree of hydrophobicity; wetting; thermal effects; moisture; moisture absorption

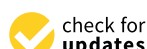



## 1. Introduction

A necessary condition for obtaining high-quality materials and products for various purposes is the control of complex physicochemical processes occurring in the contact zone during the materials structure formation [1,2].

An important component of composite materials, such as polymers, paints, industrial rubber goods, and asphalt, are fillers, which occupy most of the total volume, having a significant impact on the formation of their structures and properties [3–5].

It is known that powdered mineral materials, due to their highly developed surface and hydrophilicity, absorb moisture well from the air and then agglomerate, which limits the possibility of their use as fillers in composite materials [6]. For example, fillers' moisture reduces adhesion in the «polymer-filler» and «bitumen-mineral filler» systems. Particle agglomerates are not completely broken down during the preparation of composites, which makes the structure of the composite material heterogeneous [7]. It negatively affects the mechanical, electrical, and other properties of the filler-containing composites [8,9]. To eliminate this negative phenomenon, the hydrophobization of the filler surface is used [6].

The literature resources [10–15] demonstrate that by changing the hydrophilic–hydrophobic filler properties using physical–chemical modification, it is possible to effectively influence the processes occurring at the filler–binder interface, on which composite physical–mechanical characteristics largely depend.

For the authors of this study, this problem is most interesting from the point of view of hydrophobized mineral fillers application in the organic–mineral mixtures with the bitumen used as a binder. Carbonate fillers based on limestone and dolomite, obtained by the dry grinding of parent rocks without extra additives, are most widely used in asphalt composites [16–18]. However, hydrophobized mineral powders are better wetted with bitumen, do not absorb moisture, do not consolidate during storage and transportation, and have reduced porosity. It has a positive effect on the asphalt concrete characteristics [19–21]. It should be noted that researchers from ex-USSR are actively involved in the issues of surface modification. Researchers outside the post-Soviet space do not consider the hydrophobization of mineral powders as a way to improve the quality of bitumen–mineral composites. The global literature resources provide the separate studies' results of the hydrophobic fillers' effect on the asphalt concrete properties, but they are devoted to the use of non-ordinary fillers, for example, amorphous carbon powder [22] and hydrophobic diatoms [23]. Moreover, according to the results of numerous foreign studies, the nature of mineral powder does not significantly affect the properties of asphalt concrete [24–26].

Scientific publications analysis showed that most of the studies on the mineral powders' hydrophobization problem for organic–mineral mixtures are of an applicable nature. There are not so many works devoted to the fundamental side of this problem, focused on the study of the mechanisms of the changes occurring and the wetting conditions as a key factor determining all other properties [27–30]. However, even in them, the wetting conditions are characterized only by the contact angle [31]. In most other studies, the hydrophobicity of powders was assessed visually [21,32–34]. But in recent years, studies have appeared in which modern methods and the assessment criteria of the mineral powders' hydrophobicity degree have been proposed [20,35,36]. At the same time, in many related fields of materials science, the wetting conditions and energy characteristics of the solids wetting are estimated from the data of adsorption tests and calorimetry [37–40]. But in the practice of road construction materials science, these methods are rarely used, especially in relation to hydrophobized materials [41–45].

In addition, when surfactants are used to modify fillers and aggregates surface, there is no clear assessment criterion of the modifier effect on the filler surface hydrophobicity degree.

Taking into account the information mentioned above, it is relevant to formulate studies of a complex fundamental-applied orientation, in which not only the use of additives is substantiated on the basis of the results of the physical and mechanical properties of composites, but also the processes occurring during the wetting of fillers are comprehensively studied. It will make it possible to purposefully approach their modification.

The study purposes were to research the limestone-based filler modification effect on the hydrophobicity degree, investigate the wetting of the surface with liquids of different polarity, and establish the relationship between the characteristics of hydrophobized mineral powders and their adsorption capacity in relation to water.

## 2. Materials and Methods

### 2.1. Materials

As a filler, the work investigated limestone powder manufactured by an aggregate processing plant, Kaluga Region, and it's chemical composition is presented in Table 1.

**Table 1.** Chemical composition of limestone powder.

| Oxides | CaO | MgO | SiO$_2$ | Al$_2$O$_3$ | Fe$_2$O$_3$ | K$_2$O | Other | LOI |
|---|---|---|---|---|---|---|---|---|
| wt.% | 51.25 | 0.88 | 3.87 | 1.65 | 0.77 | 0.14 | 0.29 | 41.15 |

The mineral composition is dominated by calcite (CaCO$_3$). Quartz is present in a small amount (SiO$_2$ content is 3.87).

As hydrophobic components, "Preparation GF-1" (GF-1) and "Preparation GF-2" (GF-2) produced by Selena limited (Shebekino, Russia) are used [46]. These are fatty acid-based supplements: GF-1 is based on a saturated acid (stearic acid); GF-2 is based on an unsaturated acid (oleic acid). To prepare the hydrophobized fillers, the limestone powder was pre-dried in an oven, then hydrophobizator was added to the hot powder and processed in a ball mill. According to the previous studies' results [21,33], the optimal production parameters were established to achieve the maximum effect: for the filler hydrophobized with 0.4 wt.% of GF-1, the processing time is 15 min (called Limestone+GF-1), and with 0.6 wt.% of GF-2, the processing time is 7 min (called Limestone+GF-2). The granulometric composition of the initial and hydrophobized limestone powders is presented in Table 2.

**Table 2.** Granulometric composition of fillers.

| Filler Type | Passage, wt.%, through a Sieve Sizes | | |
|---|---|---|---|
| | 2 mm | 0.125 mm | 0.063 mm |
| Limestone | 100 | 99 | 90 |
| Limestone+GF-1 | 100 | 100 | 92 |
| Limestone+GF-2 | 100 | 100 | 93 |
| Requirements of Russian Standard GOST R 52129-2003 | No less 100 | No less 85 | No less 70 |

### 2.2. Methods

The dispersity of the fillers was assessed by the particle size distribution obtained by laser diffractometry using an Analysette 22 NanoTec plus particle size analyzer (Fritsch GmbH, Idar-Oberstein, Germany). The sample preparation involved preparing suspensions of limestone powders in distilled water. For the fillers hydrophobized with surfactant, the analysis was carried out with the ultrasonication of the suspension.

The microstructural features of the fillers were studied using a MIRA 3 LMU scanning electron microscope (TESCAN, Brno, Czech Republic). Before SEM analysis, the particles of the fillers were coated with chromium using sputtering method.

The degree of hydrophobicity of the limestone-based fillers was determined according to the procedure depicted in Figure 1. Distilled water was poured into a glass beaker, and a weighed portion of a limestone powder sample weighing 0.100 ± 0.001 g was poured. The weight of the sample was selected based on achieving the most uniform distribution of particles over the water surface with the thinnest layer, and minimizing the aggregating of hydrophobic particles, which greatly affect the results. The beaker was placed in a rotation shaker and stirred for 1 h at 140 ± 10 rpm. After the end of stirring, the solution was kept at rest, after which the hydrophobic part was collected from the water surface with filter paper until the surface was completely cleaned. The volume of water remaining in the glass with the settled powder particles was filtered through a paper filter. Next, the percentage of wetted particles removed by the filter and the content of hydrophobic

particles were calculated. The final result of the degree of hydrophobicity was the average of five measurements.

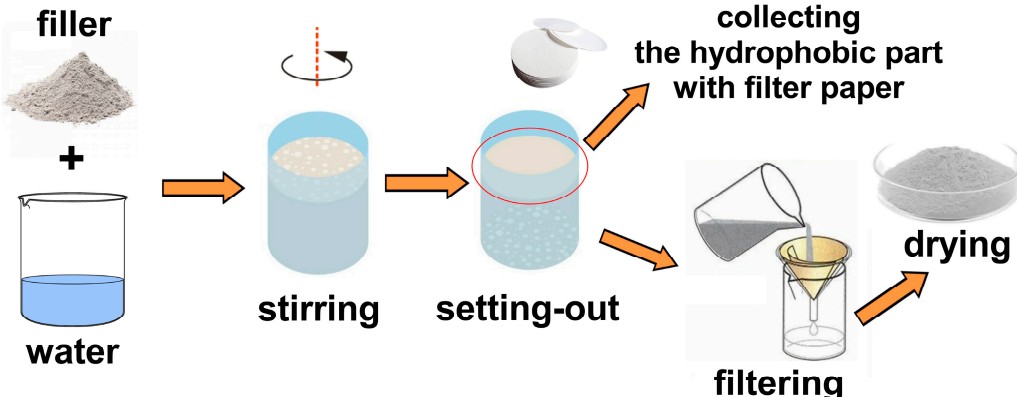

**Figure 1.** Scheme of the method for determining the degree of hydrophobicity.

In the present study, the direct measurements of the contact angles were carried out for prepared double-layer samples obtained by pressing a powder (top layer) in a mixture with boric acid (samples base—bottom layer) using a DSA30 device (KRÜSS GmbH, Hamburg, Germany) according to the sessile drop method. The contact angle value for each composition was calculated as the average of six measurements (droplets).

The kinetics of heat release were measured using a ToniCAL model 7338 differential heat flow calorimeter (Toni Technik Baustoffprüfsysteme GmbH, Berlin, Germany). The operating principle of the calorimeter is based on measuring the difference in the heat fluxes between the ampoule with the test sample and the ampoule with the reference one. In the current experiment, the three following test liquids were used: polar—distilled water, monopolar—benzene, and non-polar—n-heptane. The energy parameters of liquids are presented in Table 3 [47].

**Table 3.** Components of the test liquids' surface tension used in the heat release kinetics measurements [47].

| Test Liquid | Surface Energy, $\gamma$, mJ/m$^2$ | | | | |
|---|---|---|---|---|---|
| | Total Surface Energy $\gamma^{total}$ | Dispersive (Lifshitz-van der Waals) Component $\gamma^{LW}$ | Polar (Acid–Alkaline) Component $\gamma^{AB}$ | Acid Component $\gamma^{+}$ | Alkaline Component $\gamma^{-}$ |
| distilled water | 72.8 | 21.8 | 51.0 | 25.5 | 25.5 |
| benzene | 28.9 | 28.9 | $\approx 0$ | $\approx 0$ | 0.96 |
| n-heptane | 20.1 | 20.1 | 0 | 0 | 0 |

From the thermodynamic data, benzene, despite the surface energy polar component zero value, is considered predominantly monopolar, since there is a small alkaline component of energy [47].

The measurements were carried out as follows. A powder weighted portion (5–10 g) was poured into a test tube, which was placed in a measuring cylinder located in the calorimeter measuring cell. For wetting, a test liquid was placed in the syringe above the sample in an amount twice the mass of the powder. After reaching thermal equilibrium inside the measuring cell, the solid and liquid substances were mixed under pressure for a few seconds. The heat energy released immediately after mixing the components and during a certain measurement period was recorded as thermal electromotive force (EMF) (mV). The device software ToniDCA allows the real-time determination of the heat release rate as a function of time (J/(g × h)). The result is displayed in the form of a differential (J/(g × h)) and an integral (J/g) curves of the released amount of heat and tabular data.

The essence of the moisture absorption determining method was as follows. Samples of the pre-dried hydrophobized and reference fillers were weighed in sample bottles on an analytical balance and placed in a desiccator filled with distilled water on a porcelain insert with holes for air circulation, located above the water surface. Weighing was carried out at regular intervals until complete saturation, and moisture absorption was calculated.

The moisture value was determined according to the standard method and as the ratio of the mass of moisture removed from the powder to the mass of dry powder.

The moisture absorption and humidity values were the average of five measurements.

## 3. Results

### 3.1. Effect of Hydrophobizing Treatment of Fillers on the Dispersion and Morphology of Limestone-Based Particles

The granulometry, together with the shape and surface morphology of mineral particles, are important characteristics of fillers in composite materials. Their assessment is the first step for correlating moisture and moisture adsorption results, and explanations of occurring phenomena as well.

A comparison of the obtained particle distribution data showed that after treatment with hydrophobic components, the peak of the size distribution shifts to the left, to the region of smaller sizes: 14.26 μm for limestone MP and 11 μm for limestone fillers modified with GF-1 and GF-2 (Figure 2a). At the same time, the overall range of particle size distribution remains approximately the same—from 100 nm to 100 μm. The increase in the degree of dispersion is clearly visible in the accumulative distribution curves which lie to the left for hydrophobized powders (Figure 2b). The detected change in the granulometry of the fillers during hydrophobization can be explained by the hydrophobization technology: the initial limestone powder was subjected to short mechanochemical treatment in a mill with GF-1 and GF-2. As a result, additional grinding and an increase in the degree of dispersion occurred.

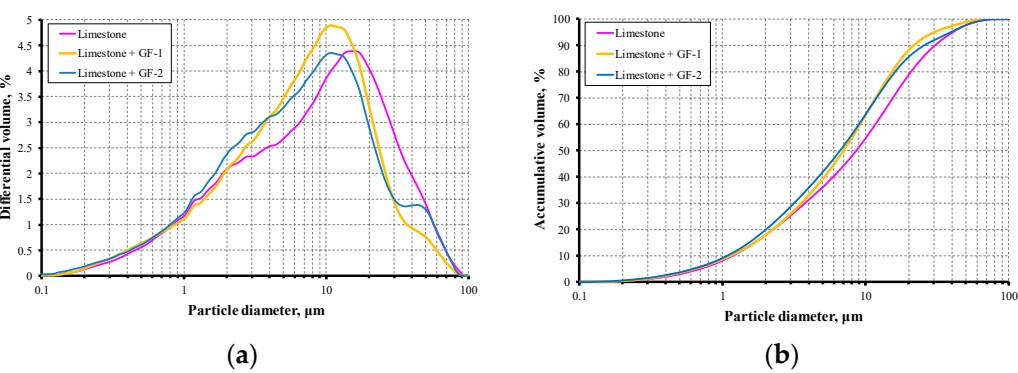

(**a**)                                                                                    (**b**)

**Figure 2.** Differential (**a**) and accumulative (**b**) particle size distribution curves of limestone-based fillers before and after hydrophobization.

The study of the microstructural characteristics of fillers using SEM analysis did not show significant changes in the shape and morphology of the particle surface as a result of hydrophobization (Figure 3). A slight agglomeration of the smallest particles less than 1 μm in size is visible both on the untreated filler (Figure 3a) and on the hydrophobized ones (Figure 3b,c). In the original limestone particles, this is due to the hydrophilicity of the surface of highly active submicron particles. After hydrophobization, the particles begin to assemble into aggregates, probably due to hydrophobic interactions. This phenomenon is described in Section 3.3. In any case, the contribution of submicron particles to the processes of moisture and moisture adsorption described in this study will be low and approximately the same for all fillers, since the proportion of these particles is less than 10 vol.% (Figure 2b).

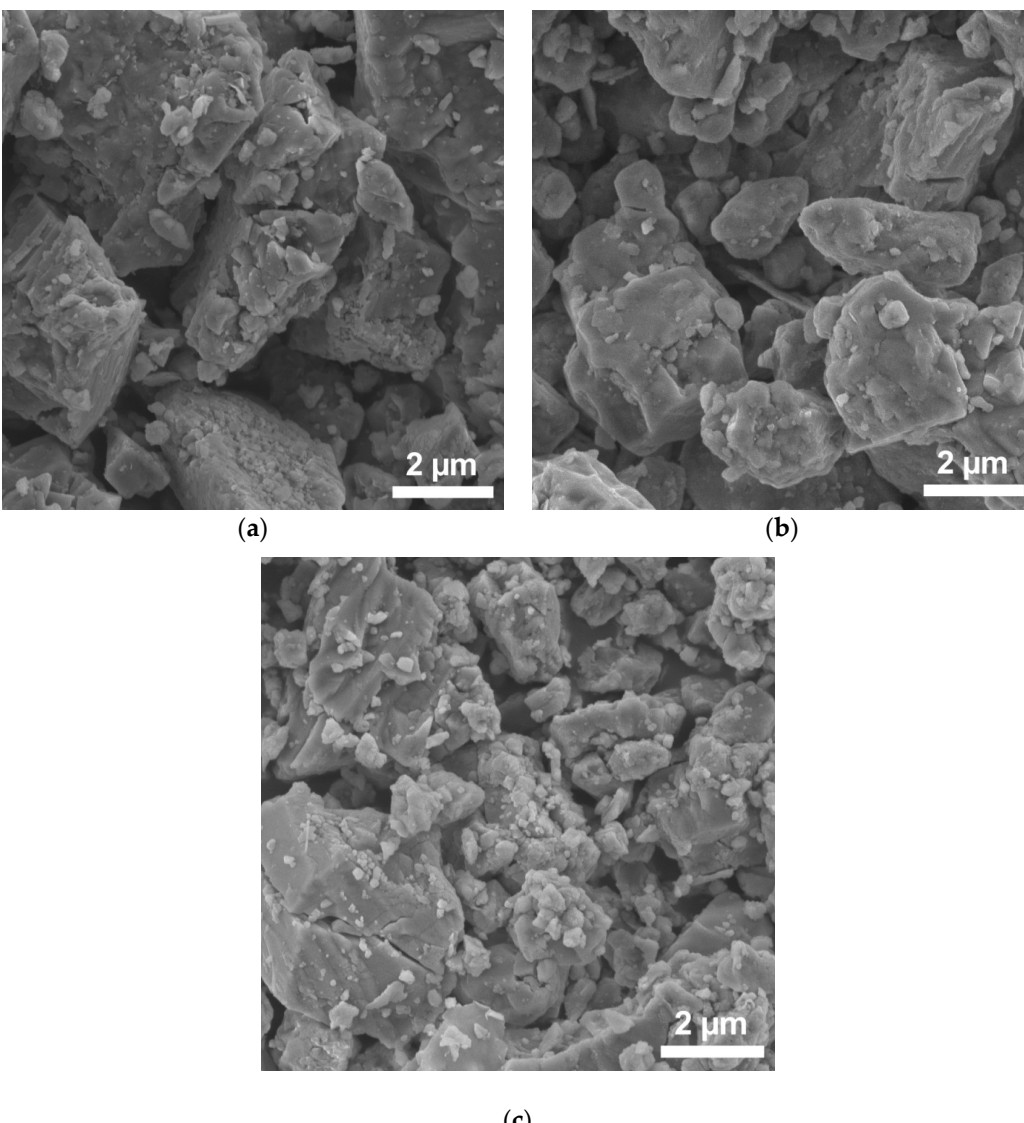

**Figure 3.** SEM images of limestone fillers before (**a**) and after hydrophobization with GF-1 (**b**) and GF-2 (**c**).

The results obtained show that hydrophobizing treatment of limestone-based fillers does not lead to significant changes in the particle size distribution and surface morphology of the particles. The fillers have approximately the same dispersion. A slight increase in grinding fineness during hydrophobization as a result of additional grinding does not have a large impact on the physical characteristics measured in the manuscript.

### 3.2. Mineral Powders Hydrophobicity Assessment

According to Russian and foreign sources, the hydrophobicity of the dispersed materials is assessed by the contact angle [22,31,48,49], visually by the presence or absence of sediment in the water layer or wetting traces after the mechanical effect [21,32–34,50]. However, nowadays, tools for the hydrophobicity degree to quantitatively characterize are needed, for example, by the percentage of particles with hydrophobic properties in the material [48].

The hydrophobicity degree determining the results by the content of the hydrophobic component in limestone powder is presented in Table 4.

**Table 4.** Modified limestone fillers' hydrophobicity degree.

| Filler Type | Content of Hydrophobized Particles, % |
|---|---|
| Limestone | 0 |
| Limestone+GF-1 | 58.2 |
| Limestone+GF-2 | 85.9 |

According to the results obtained, the treatment of limestone powders with modifiers gives a different content of hydrophobized particles. The filler modified with GF-2 is practically not wetted with water (the hydrophobicity degree is more than 85%), and the treatment with GF-1 makes the limestone powder hydrophobic a little more than half (the hydrophobicity degree is about 58%). The differences in the results can be explained by the different composition and action mechanism of the hydrophobic components. In particular, this may be due to the different chemical basis of the hydrophobic agent: unsaturated oleic acid in GF-2 has a shorter molecular length than stearic acid in GF-1, which is explained by the presence of a double bond in the structure and rotation of the hydrocarbon radical towards the carboxyl group, i.e., to the surface [51]. As a result, the hydrocarbon chain is less mobile than the chain of the stearic acid molecule and cannot curl into a ball. Thanks to this, the hydrophobic agent based on oleic acid adheres more tightly to the surface and provides better hydrophobization [52].

Previous studies [33] established the hydrophobicity of the filler treated with GF-1 by the free-floating method. However, the results of this static test do not make it possible to assess the actual content of the hydrophobized particles in limestone powder. Therefore, the technique used in this research gives significantly more reliable results in quantitative terms, which makes it possible to substantiate the effectiveness of the choice of a hydrophobic component, its concentration, and the technology for obtaining modified fillers.

Also, the well-known method for assessing hydrophobicity by the contact angle of wetting with water was used, which allows the indirect assessment of the energy state of the surface of a solid.

In the case of an untreated powder, the surface is wetted and the liquid is filtered into the pores; as a result, the final contact angle is not formed. The result was recorded in the video file, from which a photograph was created (Figure 4a). Since the limestone powder is hydrophilic, the value of the contact angle of wetting is varied from 0 to 90°.

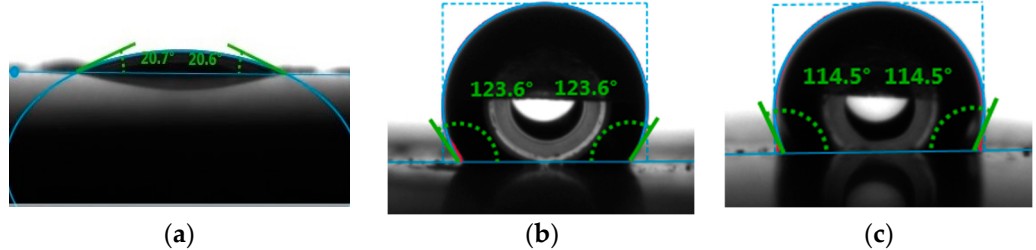

(**a**)            (**b**)            (**c**)

**Figure 4.** Distilled water drops on the limestone-based filler surface before (**a**) and after treatment with GF-1 (**b**) and GF-2 (**c**).

After hydrophobization, a final contact angle that formed on the sample surface was ≈123.6° for the Limestone+GF-1 composition (Figure 4b) and ≈114.5° for the Limestone+GF-2 composition (Figure 4c).

This indicates that the treatment has achieved its goal, and the particles of the limestone powder are no longer wetted with water.

It is noteworthy that the results of contact angles measuring and determining the hydrophobicity degree do not completely correlate with each other. However, during the observations, it was noted that the contact angle on the powder modified with GF-1 decreases over time. This occurs despite the fact that in the first few minutes, when the photographs were obtained (Figure 4b), the final contact angle was formed. Subsequently,

part of the filler grains is becoming wetted probably, and the shape of the droplet changes. There was no such thing on the limestone powder modified with GF-2.

### 3.3. Thermal Effects of Powder Wetting with Different Polarity Liquids

The microcalorimetry application in this study aims to quantify the limestone surface wetting before and after hydrophobization through the enthalpy of wetting by liquids of different polarity. When the initial limestone powder is wetted with water, heat is released (Figure 5a).

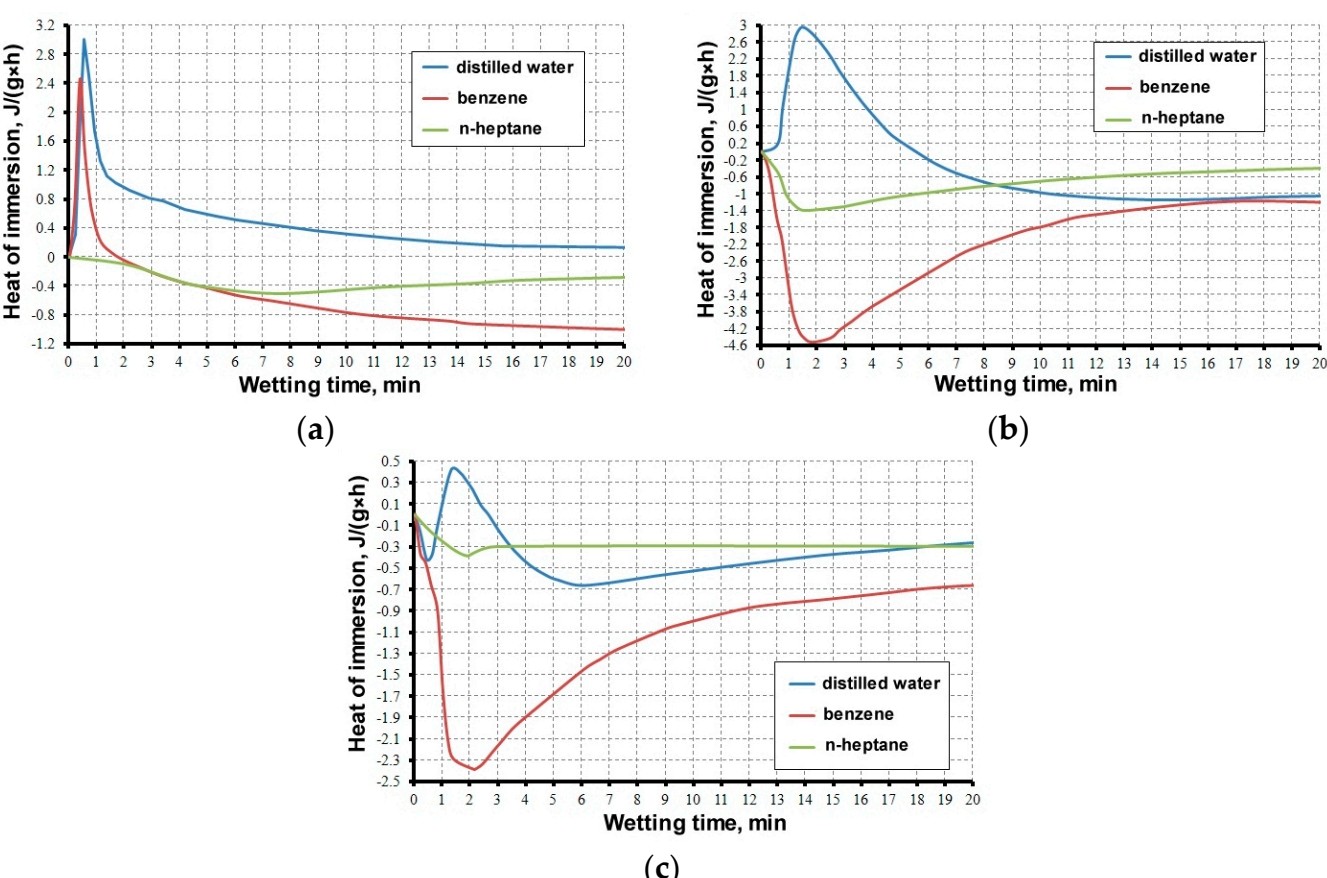

**Figure 5.** Intensity of heat release/heat absorption upon wetting of limestone powder with distilled water, benzene, and n-heptane before (**a**) and after hydrophobization with GF-1 (**b**) and GF-2 (**c**).

The total wetting time was 1.5 h. The maximum release/absorption of heat during wetting is observed in the first minutes of liquid introduction. The character of the wetting curve with benzene is similar to water, but the position of the maximum differs—for water: 3.01 J/(g × h) occurs at 35 s, for benzene: 2.45 J/(g × h)—at 24 s. It is very important that after ≈1.5 min, the heat of the immersion curve goes into the region of negative values and reaches a value of 1.07 J/(g × h) at 20–25 min, after which it remains more or less constant for the entire measurement time. In the case of n-heptane, only an endothermic effect of lower intensity is observed than in the case of benzene.

To explain the obtained effects, we will assume that the process of wetting a solid with a liquid is the closest to the process of dissolution. The heat effect or the change in enthalpy upon dissolution according to Hess's law should be considered as the sum of the thermal effects of physical and chemical processes. The physical process is associated with the distribution of substance particles between solvent molecules and is accompanied by heat absorption [53]. In the case of a hydrophilic filler, this occurs when wetting with non-polar and low-polar liquids—n-heptane and benzene. A chemical process involves the interaction of liquid molecules with particles of the introduced substance, as a result of

which new bonds or compounds are formed [51]. In this case, heat is released [53]. This effect is observed when hydrophilic limestone is wetted with a polar liquid—distilled water. The presence of a slight exothermic effect when the chemically active initial limestone is wetted with benzene can be associated with the presence of π-electrons in the benzene ring, which impart activity to it, including with respect to the active functional groups of a certain acidity on the surface of the carbonate material [54].

The character of the heat release curves after hydrophobization changes significantly (Figure 5b,c). Only n-heptane repeats dependence, established early—the wetting process is accompanied by heat absorption, while the extremum occurs much earlier—after 1.5–2 min. This circumstance can presumably be interpreted as an improvement in the wetting of mineral powders after hydrophobization with a non-polar liquid, which is very important for the interaction of mineral powders with organic binders.

When the treated fillers are wetted with benzene, an endothermic effect takes place with a pronounced minimum, which occurs after ≈2 min (Figure 5b,c). The differential values of the absorbed heat significantly exceed the quantitative indicators on the original limestone: $-4.53$ J/(g × h) for the Limestone+GF-1 composition and $-2.39$ J/(g × h) for the Limestone+GF-2 composition. In this case, a very small number of active centers or their complete absence on the surface during the adsorption of the hydrophobic component provides to the fact that a certain activity of benzene is not appearing. Wetting takes place with a non-polar liquid with a higher surface tension than for n-heptane, which determines the greater magnitude of the heat effects' maximum that occurs earlier than in the case of a hydrophilic filler, which may indicate wetting improvement.

The greatest changes occurred in the curves' character via distilled water application. In particular, there is a shift in the maximum of the differential enthalpy towards an increase in time (from 35 s to ≈1.5 min) and its general decrease (from 3.01 to 2.96 J/(g × h) for the material treated with GF-1, and 0.42 J/(g × h) for the material treated with GF-2) (Figure 5b,c). Further, after ≈3 min for the Limestone+GF-2 composition and ≈5.5 min for the Limestone+GF-1 composition, the exothermic effect ends, and heat absorption begins (Figure 5b,c). The absolute values of the endothermic effect are significantly lower than in the case of using benzene, since, probably, water has a lower value of the dispersive component of the surface tension. In this case, there is no visual wetting of the surface of the compressed powder, and a hydrophobic effect is observed (Figure 4b,c).

Based on the results mentioned above, the exothermic effect occurs when limestone powder is wetted with water both before and after hydrophobization (Table 5).

**Table 5.** Integral values of exothermic effects when limestone powders are wetted with liquids of different polarity.

| Test Liquid | $\gamma^{polar}$ mJ/m$^2$ | $\gamma^+$, mJ/m$^2$ | $\gamma^-$, mJ/m$^2$ | Integral Effect When Wetting the Powder, mJ/m$^2$ | | |
|---|---|---|---|---|---|---|
| | | | | Limestone | Limestone+GF-1 | Limestone+GF-2 |
| distilled water | 51.0 | 25.5 | 25.5 | 100 | 63 | 6 |
| benzene | ≈0 | ≈0 | 0.96 | 5 | 0 | 0 |
| n-heptane | 0 | 0 | 0 | 0 | 0 | 0 |

In the order of decreasing the values of the total heat of immersion, the fillers can be arranged in the following sequence: "limestone" → "limestone+GF-1" → "limestone+GF-2". Probably, that materials are located at mentioned sequences from the point of view of the chemical interactions' occurrence at the interface with distilled water—a polar liquid. The initial limestone powder is the most active, since there are active adsorption centers on its surface, on which interaction with water ions H$^+$ and OH$^-$ occurs. Other researchers also note a significant exothermic effect, and not only for carbonate materials [54–58].

The explanation for the predominantly physical interactions' occurrence at the interface between polar water and hydrophobized fillers can be a very small amount of active centers on the surface as a result of the additives' adsorption in the main part of

them, which reduces the probability of chemical interaction. As a result, the so-called "hydrophobic interactions" take place, which appear in aqueous solutions as a result of the interaction of water molecules with non-polar hydrophobic particles. The thermodynamic disadvantage of the water molecules' contact with non-polar surfaces causes a strong attraction of the molecules to each other. The injection of hydrophobized limestone-based particles into water disrupts the spatial structure of the strong hydrogen bonds formed by water molecules, for which energy is expended and heat is absorbed [59–61].

The nature of the interaction and the sign of the thermal effect during combining carbonate powder before and after hydrophobization with liquids of different polarity are confirmed by the results of other studies [52,62–64].

The noted differences in the nature of heat release and heat absorption in modified fillers indicate significant diversity in the composition and action mechanism of both hydrophobic components, despite the fact that effective hydrophobization was achieved due to using both ones.

### 3.4. Changes in the Properties of Fillers after Hydrophobization

The key properties of fillers during the research of hydrophobization processes should take into account those processes characterizing the interaction with water. In the present work, the moisture and moisture absorption of the studied limestone powders were determined (Figure 6).

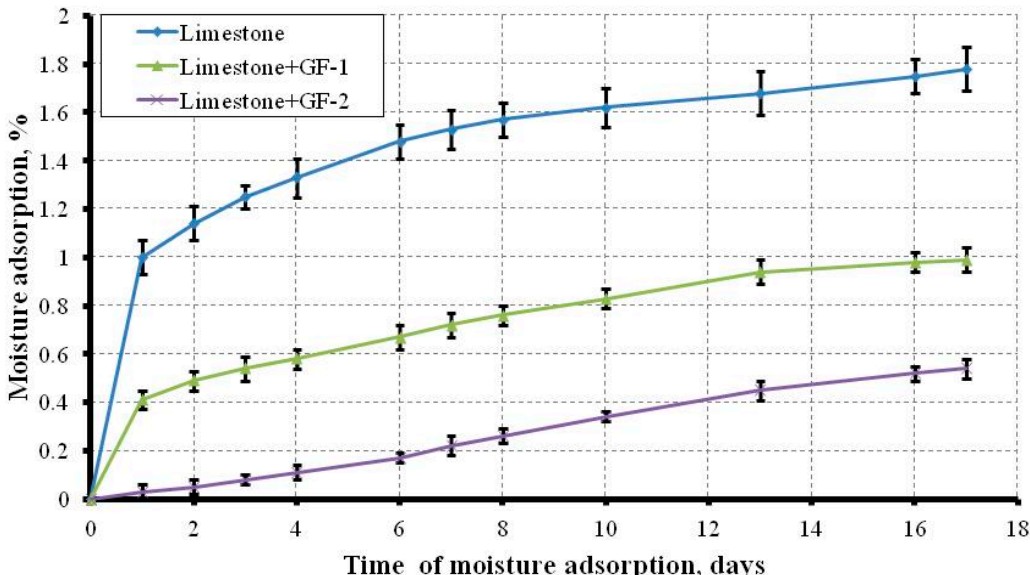

**Figure 6.** Change in moisture absorption of limestone powders before and after hydrophobization over time.

As expected, the hydrophobization of the limestone powder results in a significant decrease in its adsorption capacity with respect to water (Figure 6). The character of the moisture absorption curves of the filler treated with GF-1 and the initial powder is similar. With increasing time, the difference between them increases from 0.6% to 0.8%. This behavior can be explained by the sufficiently large amount of hydrophilic particles in the modified limestone powder. The more hydrophobic mineral powder processed with the GF-2 has a completely different moisture absorption curve: moisture adsorption occurs evenly with increasing time (Figure 6). The limiting value of moisture absorption in comparison with the initial value decreased by more than three times, which should have a positive effect on the properties of composite materials based on the studied filler.

Moreover, the limiting values of moisture absorption for fillers treated with various modifiers are in good agreement with the results obtained in terms of the hydrophobicity degree (Figure 7). Accordingly, the difference in the values of moisture absorption between

the modified and untreated powder correlates well with the percentage of hydrophobic particles in them. This logical conclusion was confirmed experimentally.

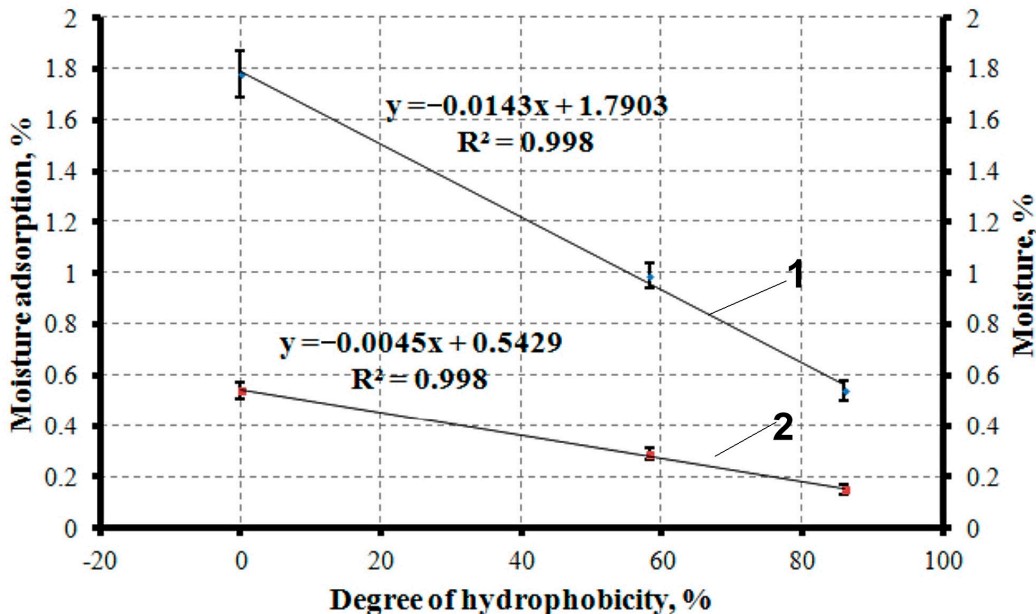

**Figure 7.** The relationship between the degree of hydrophobicity and indicators of moisture absorption (1) and moisture (2).

The limiting values of moisture absorption fully correlate not only with the degree of hydrophobicity, but also with the values of moisture determined independently of the first experiment, based on very close determination coefficients $R^2$ (Figure 7). The moisture content of the fillers hydrophobized with GF-1 and GF-2 decreased from 0.54% for the original limestone to 0.29% and 0.15%, respectively.

Thus, a linear dependence of the indicators of moisture absorption and moisture on the degree of hydrophobicity has been established. The quantitative method for assessing hydrophobicity, proposed in the article, makes it possible to obtain significantly more reliable results in comparison with the analogs used with the minimization of the effect of the hydrophobic interactions between particles (their adhesion into aggregates). This makes it possible to use the technique as a simple tool for assessing the hydrophilic–hydrophobic properties of mineral powders and, possibly, establishing the degree of their influence on the basic properties of bitumen–mineral composites.

## 4. Conclusions

Studies of a complex fundamental-applied orientation have been carried out, in which the processes occurring during the wetting of fillers have been comprehensively studied, which allows us to substantiate the use of the components GF-1 and GF-2 as effective hydrophobizers of carbonate fillers.

Through the mechanochemical treatment of the powder with the investigated hydrophobic components, it was possible to obtain fillers with a sufficiently high content of hydrophobic particles (58.2% and 85.9%, respectively). It was revealed that after treatment with hydrophobizators, the contact angles of wetting increase to 123.6° and 114.5°, respectively. These results and indicators of the degree of hydrophobicity are not entirely correlated with each other. However, it was noticed that the contact angle on the powder modified with the GF-1 decreases with time.

Investigations of the powder wetting thermal effects with different polarity liquids by microcalorimetry made it possible to establish that with an increase in the degree of the hydrophobicity of the filler, the integral heat of immersion decreases due to a significant decrease in the probability of the chemical interactions between water and

powder, due to the adsorption of the molecules of the used hydrophobizators on the active centers of limestone. The revealed endothermic effects indicate the occurrence of physical interactions due to non-polar dispersion forces. Differences in the nature of heat release and heat absorption in modified fillers indicate significant differences in the composition and mechanism of action of the used hydrophobizators, which affected the efficiency of hydrophobization.

A linear dependence of the indicators of moisture absorption and the moisture of fillers, determined by independent experiments, on the degree of their hydrophobicity has been established. The limiting value of the moisture absorption of the powder hydrophobized by GF-2, in comparison with the initial one, decreased by more than three times, which should have a positive effect on the properties of composite materials with its use.

Based on the results of a fundamental nature obtained in the manuscript, further studies will present research on the development of asphalt composites, which will analyze the influence of the proposed treatment of carbonate-based fillers on the properties of asphalt concrete. In particular, the relationships between wetting characteristics (wetting heat, contact angle, degree of hydrophobicity, etc.) and the physical and mechanical properties of asphalt composites will be established. The main focus will be on the bitumen content of the mixture, as well as on water saturation, water resistance, and volumetric stability at water saturation, i.e., properties directly or indirectly characterizing the adhesion strength between the bitumen film and the surface of mineral particles.

**Author Contributions:** Conceptualization, M.L. and V.Y.; methodology, M.L.; validation, A.A.; investigation, M.L. and E.K.; resources, V.Y.; data curation, M.L. and M.K.; writing—original draft preparation, M.L. and V.Y.; writing—review and editing, M.L., V.Y. and M.K.; visualization, M.L. and A.A.; supervision, M.L. and V.Y. All authors have read and agreed to the published version of the manuscript.

**Funding:** This research was carried out as part of the implementation of the Development Program of the flagship university on the basis of BSTU named after V.G. Shukhov.

**Data Availability Statement:** Data are contained within the article.

**Acknowledgments:** The work was realized using equipment of High Technology Center at BSTU named after V.G. Shukhov.

**Conflicts of Interest:** The authors declare no conflict of interest. The funders had no role in the design of the study; in the collection, analyses, or interpretation of data; in the writing of the manuscript; or in the decision to publish the results.

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
