# Peer review of "Hydrophilic–Hydrophobic Properties of the Surface of Modified Carbonate Fillers for Asphalt"

_jcs, doi:10.3390/jcs7120507_

Round 1

Reviewer 1 Report

Comments and Suggestions for Authors

This manuscript explores the hydrophobic characteristics of limestone powder using various hydrophobic agents, including GF-1 and GF-2. Here are my comments:

1.     In the introduction, explicitly stating the existing challenges or limitations in current carbonate filler modification would help underscore the necessity for further research and highlight the specific contributions of this study.

2.     In addition, it will be helpful if the authors clearly articulate the significance of this study and explain why the study of hydrophobized mineral fillers and their application with bitumen is important, especially in contrast to previous research.

3.     Please replace all commas in decimals with decimal points.

4.     Can the authors provide information regarding the particle distribution employed in this study? Does it have an impact on the properties?

5.     Regarding Table 4, can the authors provide additional details explaining why GF-2 significantly increases the content of hydrophobic particles?

6.     It would be beneficial to clarify how the findings of this research will be applied in the bitumen-mineral composites.

Comments on the Quality of English Language

This article would greatly benefit from concise and thorough editing

Author Response

Dear reviewer, thank you for your time and consideration. We truly value and all the comments and suggestions you have provided. The authors of this manuscript did their best to address your suggestions and made considerable text revision when it was required. 

Comment: In the introduction, explicitly stating the existing challenges or limitations in current carbonate filler modification would help underscore the necessity for further research and highlight the specific contributions of this study.

Response: Thanks for the comment. The required information has been added to the Introduction and Conclusion sections

Comment: In addition, it will be helpful if the authors clearly articulate the significance of this study and explain why the study of hydrophobized mineral fillers and their application with bitumen is important, especially in contrast to previous research.

Response: Thanks for the comment. The required information is presented in the manuscript as a following: Scientific publications analysis showed that the most of the studies on the mineral powders hydrophobization problem for organic-mineral mixtures are of an applicable nature. There are not so many works devoted to the fundamental side of this problem, focused on the study of the mechanisms of the changes occurring and the wetting conditions as a key factor determining all other properties [27–30]. However, even in them, the wetting conditions are characterized only by the contact angle [31]. In the most other studies, the hydrophobicity of powders was assessed visually [21, 32–34]. But in recent years, studies have appeared in which modern methods and assessment criteria of the mineral powders hydrophobicity degree have been proposed [20, 35, 36]. At the same time, in many related fields of materials science, the wetting conditions and the energy characteristics of the solids wetting are estimated from the data of adsorption tests and calorimetry [37–40]. But in the practice of road construction materials science, these methods are rarely used, especially in relation to hydrophobized materials [41–45].

Comment: Please replace all commas in decimals with decimal points.

Response: Thanks for the comment. Correction was done

Comment: Can the authors provide information regarding the particle distribution employed in this study? Does it have an impact on the properties?

Response: Thanks for the comment. The fillers have approximately the same dispersion. A slight increase in the degree of dispersion during hydrophobization as a result of additional grinding does not have a large effect on the properties described in the article. Therefore, an analysis of the dispersion effect was not initially provided in this study. This information has been added to the manuscript

Comment: Regarding Table 4, can the authors provide additional details explaining why GF-2 significantly increases the content of hydrophobic particles?

Response: Thanks for the comment. The required information has been added to the manuscript

Comment: It would be beneficial to clarify how the findings of this research will be applied in the bitumen-mineral composites.

Response: Thanks for the comment. The required information has been added to the Conclusion section

Reviewer 2 Report

Comments and Suggestions for Authors

The manuscript submitted by Lebedev et al. shows a comparison of two hydrophobization treatments (unknown) on limestone to be used in asphalt. Although the data is interesting, there is room for improving the work before it is ready to be accepted in the journal. Please find my point-by-point comments below.

- Please provide the percentages as wt% or vol%.

- How does this work lead to novel research and future work? Please reflect on this and clearly state it in the revised version.

- Several references are missing in the introduction. Please provide references for each statement in the introduction and within the text.

- How can the study be replicated or interpreted by the reader if at least some basic information regarding the hydrophobization treatment is not presented? Even if the information is protected, a general statement should be provided on what type of treatment this refers to. This is even more important for comparison purposes.

- How was the protocol to determine the degree of hydrophobicity established? Also, it is rather complex. Therefore, a process scheme should be included to simplify this for readers.

- How reproducible are the results from the different and unknown hydrophobization treatments in this study? Can the materials be repeatedly exposed without losing hydrophobicity?

- What is the surface morphology of the hydrophobic particles? This is critical to correlate macroscopic properties (such as contact angle) with the material microstructure and the different hydrophobic phenomena occurring in the material.

- How does the contact angle change with time?

- Why Figure 3 and Figure 4 does not include error bars? Also, the manuscript should describe the data using statistical tools to ensure relevance among the data.

- How do the measurements in the manuscript correlate with relevant conditions where these particles will be exposed to, for instance, in asphalt application? 

Comments on the Quality of English Language

The manuscript should be revised for English-grammar checks.

Author Response

Dear reviewer, thank you for your time and consideration. We truly value and all the comments and suggestions you have provided. The authors of this manuscript did their best to address your suggestions and made considerable text revision when it was required.

Comment: Please provide the percentages as wt% or vol%.

Response: Thanks for the comment. The required correction was done

Comment: How does this work lead to novel research and future work? Please reflect on this and clearly state it in the revised version.

Response: Thanks for the comment. The required information has been added to the Introduction and Conclusion sections

Comment: Several references are missing in the introduction. Please provide references for each statement in the introduction and within the text.

Response: Thanks for the comment. The required information has been added to the Introduction section

Comment: How can the study be replicated or interpreted by the reader if at least some basic information regarding the hydrophobization treatment is not presented? Even if the information is protected, a general statement should be provided on what type of treatment this refers to. This is even more important for comparison purposes.

Response: Thanks for the comment. The required information has been added to the manuscript

Comment: How was the protocol to determine the degree of hydrophobicity established? Also, it is rather complex. Therefore, a process scheme should be included to simplify this for readers.

Response: Thanks for the comment. The required scheme has been added to the manuscript

Comment: How reproducible are the results from the different and unknown hydrophobization treatments in this study? Can the materials be repeatedly exposed without losing hydrophobicity?

Response: Thanks for the comment. As part of this study, 3 series of hydrophobized powders were obtained using the technology described in section 2.1 using each of the modifiers. The results of tests of these series of fillers for hydrophobicity and adsorption capacity fit into the experimental errors (standard deviations) added to the figures in the article. Therefore, the experimental results are reproducible.

The proposed hydrophobizing treatment is aimed at preventing agglomeration of the powder during movement and long-term storage, including in conditions with high air humidity, as well as improving the characteristics of asphalt materials. After combining the filler with bitumen, the powder particles in their pure form are not exposed to moisture, since they are already covered with a layer of bitumen. Although asphalt concrete is subject to repeated exposure to water, it is incorrect to consider this as an effect only on the powder. Despite this, the water resistance of asphalt using hydrophobized fillers increases. The purpose of the study was not to repeatedly expose the powders to water. But it was experimentally noted that the hydrophobization effect persists for a long time, including after a single exposure to water.

Comment: What is the surface morphology of the hydrophobic particles? This is critical to correlate macroscopic properties (such as contact angle) with the material microstructure and the different hydrophobic phenomena occurring in the material.

Response: Thanks for the comment. The required information has been added to the Materials and Methods section

Comment: How does the contact angle change with time?

Response: Thanks for the comment. During observations, it was noticed that the contact angle of the powder modified with the GF-1 additive decreases over time. This occurs despite the fact that in the first few minutes when the photographs were taken (in updated version of manuscript – Fig. 4b), the final contact angle was formed. Subsequently, some of the filler grains are probably wetted, and the shape of the drop changes. This was not observed on limestone modified with GF-2. Unfortunately, the numerical values of contact angles over time were not measured in this study.

Comment: Why Figure 3 and Figure 4 does not include error bars? Also, the manuscript should describe the data using statistical tools to ensure relevance among the data.

Response: Thanks for the comment. The required corrections were done

Comment:  How do the measurements in the manuscript correlate with relevant conditions where these particles will be exposed to, for instance, in asphalt application?

Response: Thanks for the comment. The manuscript presents studies of a complex fundamental and applied focus, in which the main focus is on the fundamental part, where the processes occurring when fillers are wetted with liquids of different polarities are comprehensively studied, which will allow for a targeted approach to their modification. Of course, wetting with organic, mostly non-polar liquids (benzene, n-heptane) differs from wetting with bitumen in conditions, because it is experimentally difficult to detect the wetting of heated fillers and bitumen at temperatures of 150–160°C. But from the point of view of chemical properties, the studies carried out in the manuscript are consistent with the liquids used. In the case of water, the conditions for the influence of water on asphalt are similar (in temperature and pressure) to the experimental ones. But after combining the filler with bitumen, the powder particles in their pure form are not exposed to moisture, since they are already covered with a layer of bitumen. Although asphalt concrete is subject to repeated exposure to water, it is incorrect to consider this as an effect only on the powder.

Reviewer 3 Report

Comments and Suggestions for Authors

This article discusses the hydrophobicity and wetting behavior of modified carbonate fillers for asphalt. The study examines the effect of filler modification on hydrophobicity, wetting with liquids of different polarity, and the relationship between hydrophobized mineral powders and their adsorption capacity. Surface-active additives are used to modify the fillers, and various tests are conducted to evaluate their hydrophobicity and wetting characteristics. The results show that the degree of hydrophobicity and wetting behavior do not always correlate, and the thermal effects of wetting indicate physical interactions due to non-polar dispersion forces. The study establishes a linear relationship between moisture absorption and the degree of hydrophobicity.

The paper is well written and well understood. My recommendation is to accept it after minor corrections:

L98: Move to the next page.

Table 2: Use as decimal separator "." and not ",".

L116: Change the sentence, put 2 periods at the end or say that it is described below (see line 142 for example).

L118: Use the correct decimal separator (same as the comment made for table 2). Revise the whole text. Also, the magnitude would be expressed as 0.100±0.001 g.

L121: Use Ctrl+Shift+Space to avoid separating the magnitude from the units.

Table 3 and 4: Set the decimal separator correctly.

Figure 2. Put n-Heptane instead of Heptane.

L213: Seconds are expressed as "s" in the international system of units. Minutes are expressed as "min". Revise the entire text.

L245: Insert n-Heptane instead of Heptane.

L252... Express minutes correctly (min).

Table 5: Use the correct decimal separator "."

Figure 4. Put 3 significant figures to R2. Put the equation in the text to be able to compare the ordinates at the origin and also the slopes of the lines. Also set the decimal separator to ".". Does the difference of slopes obtained have any implication?

Author Response

Dear reviewer, thank you for your time and consideration. We truly value and all the comments and suggestions you have provided. The authors of this manuscript did their best to address your suggestions and made considerable text revision when it was required.

Comment: L98: Move to the next page.

Response: Thanks for the comment. The required correction was done

Comment: Table 2: Use as decimal separator "." and not ",".

Response: Thanks for the comment. The required correction was done

Comment: L116: Change the sentence, put 2 periods at the end or say that it is described below (see line 142 for example).

Response: Thanks for the comment. The required correction was done

Comment: L118: Use the correct decimal separator (same as the comment made for table 2). Revise the whole text. Also, the magnitude would be expressed as 0.100±0.001 g.

Response:  Thanks for the comment. The text has been revised. The required correction was done

Comment: L121: Use Ctrl+Shift+Space to avoid separating the magnitude from the units.

Response: Thanks for the comment. The required correction was done

Comment: Table 3 and 4: Set the decimal separator correctly.

Response: Thanks for the comment. The required correction was done

Comment: Figure 2. Put n-Heptane instead of Heptane.

Response: Thanks for the comment. The required correction was done

Comment: L213: Seconds are expressed as "s" in the international system of units. Minutes are expressed as "min". Revise the entire text.

Response: Thanks for the comment. The text has been revised. The required correction was done

Comment: L245: Insert n-Heptane instead of Heptane.

Response: Thanks for the comment. The required correction was done

Comment: L252... Express minutes correctly (min).

Response: Thanks for the comment. The required correction was done

Comment: Table 5: Use the correct decimal separator "."

Response: Thanks for the comment. The required correction was done

Comment: Figure 4. Put 3 significant figures to R2. Put the equation in the text to be able to compare the ordinates at the origin and also the slopes of the lines. Also set the decimal separator to ".". Does the difference of slopes obtained have any implication?

Response: Thanks for the comment. The required correction was done. This Figure shows the correlation between the adsorption capacity of powders towards water and their degree of hydrophobicity. It turned out that the results of moisture absorption and humidity separately are well described by a linear relationship. The physical meaning of the slope of the obtained direct correlation is difficult to detect, since at 100 % hydrophobicity, humidity and moisture absorption are different from zero. Perhaps other dependencies should be used to construct the correlation curve. However, these studies were not carried out within the scope of this manuscript, as this would require a large amount of additional research work.

Round 2

Reviewer 1 Report

Comments and Suggestions for Authors

The authors answered the questions clearly. The manuscript is recommended for publishing.

Reviewer 2 Report

Comments and Suggestions for Authors

No additional comments or suggestions. 

Comments on the Quality of English Language

No additional comments on the Quality of the English, should be check by the Editorial office. 

Reviewer 3 Report

Comments and Suggestions for Authors

All my comments have been considered. I recommend the publication of this article in materials.